# Graves-PCD: protocol for a randomised, dose-finding, adaptive trial of the plasma cell-depleting agent daratumumab in severe Graves' disease

Faye Wolstenhulme [ID],[1] Irena Bibby [ID],[1] Michael Cole,[2] Lydia Grixti,[3] Naomi McGregor,[1] Penny Bradley,[4] Rebecca H Maier [ID],[1] Jenn Walker,[1] Simon H Pearce,[3] James Wason [ID] [2]

[1]Newcastle Clinical Trials Unit, Newcastle University, Newcastle upon Tyne, UK
[2]Population Health Sciences Institute, Newcastle University, Newcastle upon Tyne, UK
[3]Translational and Clinical Research Institute, Faculty of Medical Sciences, Biomedicine West, International Centre for Life, Newcastle University, Newcastle upon Tyne, UK
[4]Pharmacy Clinical Trials, Newcastle upon Tyne Hospitals NHS Foundation Trust, Newcastle upon Tyne, UK

**Correspondence to**
Professor James Wason;
james.wason@newcastle.ac.uk

## ABSTRACT

**Introduction** Severe Graves' disease is a life-changing condition with poor outcomes from currently available treatments. It is caused by directly pathogenic thyroid-stimulating hormone receptor-stimulating antibodies (TRAb), which are secreted from plasma cells. The human anti-CD38 monoclonal antibody daratumumab was developed to target plasma cells which express high levels of CD38, and is currently licensed for treatment of the plasma cell malignancy, myeloma. However, it can also deplete benign plasma cells with the potential to reduce TRAb and alter the natural history of severe Graves' disease. This study aims to establish proof of concept that daratumumab has efficacy in patients with severe Graves' disease and will provide important data to inform a choice of dosing regimen for subsequent trials.

**Methods and analysis** The Graves-PCD trial aims to determine if daratumumab modulates the humoral immune response in patients with severe Graves' disease, and if so, over what time period, and to find an optimal dose. It is a single-blinded, randomised, dose-finding, adaptive trial using four different doses of daratumumab or placebo in 30 adult patients. Part 1 of the trial is dose-finding and, following an interim analysis, in part 2, the remaining patients will be randomised between the chosen dose(s) from the interim analysis or placebo. The primary outcome is the percentage change in serum TRAb from baseline to 12 weeks.

**Ethics and dissemination** The trial received a favourable ethical opinion from London-Hampstead Research Ethics Committee (reference 21/LO/0449). The results of this trial will be disseminated at international meetings, in the peer-reviewed literature and through partner patient group newsletters and presentations at patient education events.

**Trial registration number** ISRCTN81162400.

## STRENGTHS AND LIMITATIONS OF THIS STUDY

⇒ Current treatments for patients with severe Graves' disease are unsatisfactory and there is therefore significant unmet clinical need for new treatment.
⇒ Adaptive design allows efficient investigation of multiple doses.
⇒ The long-term remission of hyperthyroidism cannot be assessed in the 24-week follow-up period.
⇒ The duration of treatment or dosing intervals will not be tested in this study.

## INTRODUCTION

Graves' disease (autoimmune hyperthyroidism) affects around 3% of women and 0.5% of men, and most commonly presents in people in their 30s and 40s with a disproportionate burden of ill-health falling on working-age women.[1 2] Typical symptoms include weight loss, palpitations, heat intolerance, tremor, breathlessness, sweating, insomnia, loss of concentration and irritability. Around 40% of patients develop thyroid eye disease (TED),[3] which can cause facial disfigurement as well as functional visual problems and even loss of sight. Quality of life in patients with TED is poor, worse than for diabetes, and similar to patients with inflammatory bowel disease.[4] Rarely, thyroid dermopathy (pretibial myxoedema) may also occur leading to a brawny thickening of the skin on the lower legs and feet.[5] These latter two problems produce particularly distressing symptoms for a relatively young and active patient group. Around 10% of patients with Graves' disease have a severe form of the disease, and experience severe thyrotoxicosis (serum FT4 $\geq$50 pmol/L (upper limit of reference range 22 pmol/L) or FT3 $\geq$15 pmol (upper limit of reference range 6.8 pmol/L)), failure of medical control of hyperthyroidism, large goitre, TED of sufficient severity to affect visual function or thyroid dermopathy. Current treatments for these patients with severe Graves' disease give unsatisfactory outcomes and are expensive, typically involving surgical thyroidectomy with or without several episodes of eye surgery.[6]

BMJ

Conventional treatment of Graves' disease with antithyroid drugs (carbimazole, methimazole or propylthiouracil) leads to remission for only around 50% of patients, decreasing to less than 20% for patients with severe disease, and does not improve eye symptoms.[7 8] Other treatment options of radioactive iodine or surgical thyroidectomy result in life-long hypothyroidism requiring levothyroxine therapy, which patients are often unwilling to accept.[9] Better treatment options are sorely needed for severe Graves' disease.

Severe Graves' disease is caused by high titres of directly pathogenic thyroid-stimulating hormone (TSH) receptor-stimulating antibodies (TRAbs), which are secreted from terminally differentiated B lymphocytes known as plasma cells.[10] Not only do TRAbs directly stimulate thyroid hormone overproduction (leading to hyperthyroidism), they also cause thyrocyte hyperplasia leading to goitre, and bind to TSH receptors on the surface of orbital fibroblasts leading to mitogenic signalling that triggers and perpetuates eye disease. Failure of medical control and/or early relapse following conventional antithyroid drugs reflects the persistence of these long-lived, TRAb-secreting plasma cells in the secondary lymphoid tissues and bone marrow.

Both benign and malignant plasma cells express high levels of the cell-surface glycoprotein CD38. Daratumumab, a biological drug that targets CD38, was recently licensed for the plasma cell malignancy, myeloma.[11 12] This human anti-CD38 monoclonal antibody has the potential to deplete plasma cells and therefore produce a rapid reduction in TRAb levels, which may alter the natural history of severe Graves' disease. This study aims to establish proof of concept that daratumumab has efficacy in patients with severe Graves' disease and will provide important data to inform a choice of dosing regimen.

Tolbert *et al* first reported the use of daratumumab in the treatment of benign, antibody-mediated haematological autoimmune disorders in 2016.[13] In this case, a female in her late teens with autoimmune haemolytic anaemia (post-stem cell transplant) was successfully

**Table 1** Objectives and outcome measures

|  | Objectives | Outcome measures |
|---|---|---|
| Primary | To determine if daratumumab modulates the humoral immune response in patients with Graves' disease | Change in serum TRAb from baseline to 12 weeks compared with change in placebo group |
| Secondary | To determine how fast daratumumab modulates the humoral immune response in patients with Graves' disease | Change in serum TRAb from baseline to 2, 4, 6, 12 and 24 weeks |
|  | To determine the optimal dose (or dose range) of daratumumab for patients with Graves' disease | Dose–response curve for daratumumab against change in serum TRAb from baseline to 6 and 12 weeks |
|  | To determine if daratumumab reduces thyroid hormone levels | Change in serum FT3 and FT4 from baseline to 2, 4, 6, 12 and 24 weeks |
|  | To determine if daratumumab changes the time course of serum TSH | Change in serum TSH from baseline to 2, 4, 6, 12 and 24 weeks |
|  | To determine if daratumumab changes thyroid size | Change in thyroid volume from baseline to 24 weeks measured by ultrasound |
|  | To determine if daratumumab changes other thyroid autoantibodies | Change in serum ATPO and thyroglobulin antibodies from baseline to 6, 12 and 24 weeks |
|  | To determine if daratumumab improves thyroid eye disease | Change in CAS, composite eye index and GOQoL score from baseline to 6, 12 and 24 weeks |
|  | To determine if daratumumab improves thyroid symptom-related QoL | Change in ThyPRO39 score from baseline to 6, 12 and 24 weeks |
|  | To determine if daratumumab is safe in this patient group | Change in serum immunoglobulins, specific antibodies including (SARS-CoV-2) and blood count parameters from baseline to 6, 12 and 24 weeks<br>Adverse reactions to 24 weeks |
| Experimental | To determine if daratumumab changes lymphocyte/plasma cell transcriptomic markers | Analysis of blood plasma cell markers and mRNA signature |
|  | To determine if daratumumab changes the lymphocyte subsets | Change in lymphocyte subsets (by FACS) from baseline to 6, 12 and 24 weeks |

ATPO, antithyroid peroxidase; CAS, Clinical Activity Score; FACS, fluorescent-activated cell sorting; GOQoL, Graves' Orbitopathy Quality of Life Instrument; QoL, quality of life; ThyPRO39, Thyroid Patient-Reported Outcome; TRAb, TSH receptor antibodies; TSH, thyroid-stimulating hormone.

treated with daratumumab and had an instant improvement in haemolysis. There are now numerous additional reports of daratumumab being repurposed to target benign disorders.[14–17]

Daratumumab is licensed for use in myeloma at a dose of 16 mg/kg; however, patients with Graves' disease have several orders of magnitude fewer plasma cells and it is therefore expected that lower doses of drug will be active in this patient group with benign disease. To address this question, the first stage of the study will be dose-finding. Notably, dose-finding trials in patients with myeloma showed no difference in adverse event (AE) rates between doses of 8 mg/kg and 16 mg/kg, and no dose-limiting toxicity up to 24 mg/kg.[11] Therefore, in order to define a signal for efficacy, this study will use 9 mg/kg as the top dose in stage 1, with reducing concentrations (3 mg/kg, 1 mg/kg and 0.5 mg/kg) along with placebo to determine the dose–response in stage 1. As myeloma trials did not find increased AEs or toxicities at higher doses, these four dose allocations will take place in parallel. Stage 2 of the study will randomise additional patients to one or more optimal doses of daratumumab, as selected from stage 1, or placebo to gain further information about real-life effect size, safety and tolerability.

## METHODS AND ANALYSIS

This study protocol is based on the Standard Protocol Items: Recommendations for Interventional Trials 2013 statement.[18]

### Sponsor and trial management

The trial sponsor is the Newcastle upon Tyne Hospitals NHS Foundation Trust (email: tnu-tr.sponsormanagement@nhs.net). Trial management is delegated to the Newcastle Clinical Trials Unit (NCTU), Newcastle University.

### Objectives

The primary objective is to determine if daratumumab modulates the humoral immune response in patients with severe Graves' disease, with the primary outcome measure the percentage change in serum TRAb from baseline to 12 weeks. Secondary objectives include determining the optimal dose of daratumumab, assessing safety, patient quality of life and impact on eye symptoms, thyroid function and goitre size (see table 1).

### Trial design

This is an adaptive, two-stage randomised phase IIa, single-centre clinical trial that will aim to recruit 30 patients with severe Graves' disease from secondary care at the Newcastle upon Tyne Hospitals NHS Foundation Trust. It is a single-blinded trial in which participants will be blind to allocation. Recruitment start date: October 2021; planned end of trial (last patient last visit): April 2024.

Stage 1 is a dose–response study using four doses of daratumumab (9 mg/kg, 3 mg/kg, 1 mg/kg, 0.5 mg/kg) and a colourless, volume-matched placebo infusion in approximately 15 patients (ie, five groups of n=3, randomised in a 1:1:1:1:1 ratio using permuted random blocks without stratification) (figure 1).

Following stage 1, an interim analysis will be performed of the first 15 patients in order to select an optimal dose(s) of daratumumab for stage 2. The dose selection will be based on an analysis of the reduction in TRAb concentration and safety assessed at 12 weeks. In stage 2, the remaining 15 patients will be randomised between placebo and one or two chosen doses of daratumumab depending on results of the interim analysis. Patients in both stages of the study will have all the same assessments and will be followed up for 24 weeks, with the primary endpoint being measured at week 12.

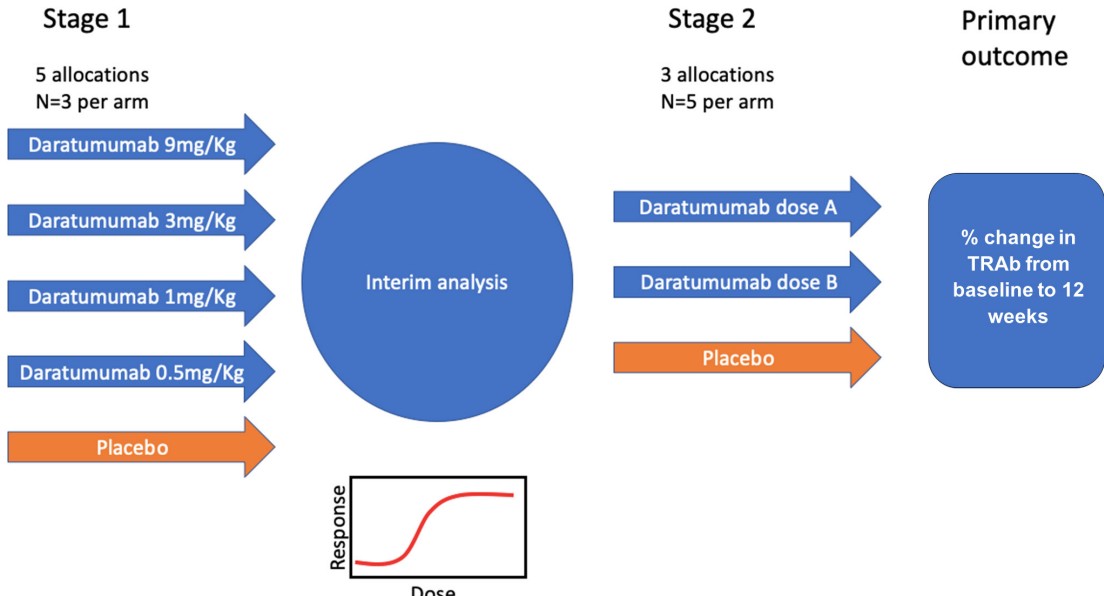

**Figure 1** Design of the study. TRAb, thyroid-stimulating hormone receptor antibodies.

## Inclusion criteria

1. Patients ≥18 years old.
2. Recent-onset Graves' disease (within 12 months) (defined as date of first thyroid function test showing hyperthyroidism (FT4 and TSH) in current episode).
3. TRAb concentrations above 10 U/L (on Roche or Brahms TRAb-binding assays).
4. One or more of:
   - Pretreatment severe hyperthyroidism (FT4 ≥50 pmol/L or FT3 ≥15 pmol/L).
   - Persisting hyperthyroidism despite more than 12 weeks of antithyroid drug therapy (defined as FT3 above the upper limit of the reference range following 12 weeks of carbimazole treatment at a dose of 40 mg or more daily (or equivalent dose of propylthiouracil)).
   - Inflammatory thyroid eye disease (defined as Clinical Activity Score ≥3) or thyroid dermopathy.
   - Large (visible) goitre (WHO grade III).
   - Two or more relapses (three episodes in total) despite completing 12 months or more of medical treatment on each occasion. Relapse is defined as FT3 above the upper limit of the local reference range.
5. For women of childbearing potential, willing to use a highly effective contraceptive method during their participation in the trial.
6. Able to understand and speak sufficient English to complete trial procedures.
7. Willing and able to provide informed consent prior to any trial procedures taking place.

## Exclusion criteria

1. Previous thyroidectomy or radioiodine treatment within 2 years.
2. Pregnant or breastfeeding, or with a plan for pregnancy within 6 months.
3. Previous shingles, known untreated cervical dysplasia, hepatitis B and C or HIV infection.
4. Anaemia (haemoglobin ≤100 g/L), thrombocytopenia (≤75×10$^9$/L) or neutropenia (≤1.0×10$^9$/L).
5. Known chronic obstructive pulmonary disease (defined as a forced expiratory volume in 1 s <60% of predicted normal), persistent asthma or a history of asthma within the last 2 years (intermittent asthma without hospitalisation is allowed).
6. Any significant physical or mental health condition that impacts the safety of the intervention, the interpretation of thyroid function or the ability of a participant to attend for intervention and safety monitoring, for example, major cardiorespiratory disease, renal or hepatic failure, pancreatitis, cancer undergoing active treatment (excluding non-melanoma skin cancer), untreated chronic infection including tuberculosis, psychosis and depression impairing activities of daily living.
7. Current use of immunosuppressive therapy for TED or other conditions (within 3 months).
8. Current or previous participation in a Clinical Trial of Investigational Medicinal Products research study within 4 months.
9. Hypersensitivity or anaphylactic reaction to previous monoclonal antibody treatments or methylprednisolone.
10. Inability, in the opinion of the investigator, to be able to complete the clinical trial visits or procedures.

## Risk assessment

Daratumumab is a licensed product for patients with myeloma and the main risk is that it will have a different side effect profile in patients with benign disease. The typical risks of daratumumab when used for myeloma include transient infusion reactions and haematological dyscrasias, the latter being accounted for by the bone marrow suppression in patients with myeloma and leading to a small risk of infection (≤5%).

Daratumumab interferes with urgent blood cross-matching and all patients will have a Coombs test and blood group and transfusion antibody screen prior to administration of daratumumab, as well as receiving an information card in case blood transfusion is needed. There is a potential for reactivation of latent viral illness, and so all participants will be screened for hepatitis B, C and HIV infection, and participants with previous shingles or current untreated cervical dysplasia will be excluded. There are no specific risks demonstrated in the Graves' disease population who will be younger and in better overall health than the patient group who currently receive daratumumab for myeloma.

Participants will be offered a course of SARS-CoV-2 vaccination prior to treatment with investigational medicinal product (IMP). If loss of immunity to specific pathogens is detected at the end of the study, the participants may be offered repeat vaccination for tetanus, pneumococcus and SARS-CoV-2, where clinically appropriate. Throughout their participation in the study, patients will continue their regular 'background' medication for treatment of thyrotoxicosis (ie, antithyroid drugs, with or without levothyroxine) as prescribed by their clinician as per usual clinical care and will be carefully monitored with regular blood tests throughout the study.

## Patient identification, consent and screening

Potential participants with severe Graves' disease within secondary care will be identified and approached in person, by post or email with a trial participant information sheet (PIS) by one of the clinical team members responsible for their usual management. Potential participants will also be identified and referred by local secondary care Patient Identification Centre sites, or can self-refer to the trial via adverts through organisations such as the British Thyroid Foundation (BTF). Consent discussions and written informed consent will take place in person at a research-specific clinic appointment, with the principal investigator or alternative member of the research team who is medically qualified and appropriately trained. For

women of childbearing potential, consent discussions will include the need to use a highly effective form of contraception (as listed on the PIS) for the duration of their participation in the trial. Consent discussions will be documented in medical notes.

Following written informed consent, eligibility for the trial will be assessed by:

▶ Demographics and medical history including medication history.

▶ Full external medical examination, including weight, heart (including resting pulse rate and 12-lead ECG), chest, abdomen, goitre size assessment, skin, eyes and neurological examination.

▶ Blood tests (full blood count (FBC), urea and electrolytes, liver function test (LFT), hepatitis B surface antigen, hepatitis C virus antibodies, HIV, bone chemistry, TRAbs, serum FT4, FT3, TSH, serum IgGAM, Coombs, blood group and red cell phenotyping).

▶ A urine sample pregnancy test for all female participants of childbearing potential (to be repeated where there is >14 days between screening and baseline visits).

The trial screening log will include reasons for screen fails and any reasons given for declining the trial.

### Randomisation
Randomisation will be performed by delegated and trained members of the site research team using the online system 'Sealed Envelope'—a central, secure, 24-hour web-based randomisation system. In both stages, the allocation sequence will be computer generated using a random permuted block design. Patients will be randomly allocated equally to the available daratumumab doses or to a matched placebo; for example, if two doses of daratumumab plus placebo are to be used, the allocation would be 1:1:1. There are no stratification factors.

In stage 1, patients will be allocated to receive a dose of daratumumab (9, 3, 1, 0.5 mg/kg) or matched placebo. In stage 2, patients will be allocated to receive daratumumab (one or two dose levels as determined by the interim analysis) or matched placebo.

### Intervention
Each participant will receive two infusions, on day 0 and day 14 (±3 days). The prescribed daratumumab dose will be prepared in a 500 or 1000 mL bag of sodium chloride 0.9%, using the most recent available participant weight to calculate banded dose (see table 2 for volume and infusion rate). Each participant will be prescribed the same banded dose for both infusions, using the same weight for both. Where placebo is allocated, a volume-matched bag of sodium chloride 0.9% will be used. All participants will be pre-medicated with three non-IMPs, paracetamol (1 g orally), methylprednisolone (100 mg intravenously) and chlorphenamine (10 mg intravenously) at least 30 min prior to their allocated infusion.

Vital signs (pulse rate, blood pressure and temperature) will be recorded prior to the start of each infusion

**Table 2** Infusion rate table

| Time (hour:min) | Rate (mL/hour) | Total volume at end of time (mL) |
|---|---|---|
| Stage 1 (1000 mL) | | |
| 1 | 80 | 80 |
| 2 | 160 | 240 |
| 3 | 240 | 480 |
| 4 | 320 | 800 |
| 4:40 | 320 | 1000 |
| Stage 2 rate A (1000 mL) | | |
| 1 | 50 | 50 |
| 2 | 100 | 150 |
| 3 | 150 | 300 |
| 4 | 200 | 500 |
| 5 | 200 | 700 |
| 6 | 200 | 900 |
| 6:30 | 200 | 1000 |
| Stage 2 rate B (500 mL) | | |
| 1 | 50 | 50 |
| 2 | 100 | 150 |
| 3 | 150 | 300 |
| 4 | 200 | 500 |

The maximum infusion rate of daratumumab was determined by the recommendation of the SmPC for daratumumab (DARZALEX 20 mg/mL concentrate for solution for infusion). For stage 1, all participants will receive the IMP/placebo diluted in 1000 mL sodium chloride 0.9% for both infusions. For stage 2, some patients may receive higher doses of daratumumab, so a more prolonged first infusion will be given to all participants. If participants have no IRRs during their first dose in stage 2, then the second dose may be given in 500 mL sodium chloride 0.9% with a faster infusion rate, in line with SmPC recommendations. IMP, investigational medicinal product; IRRs, infusion-related reactions; SmPC, Summary of Product Characteristics.

and every 15 min during the course of the infusion, and for 1 hour afterwards. Infusion-related reactions will be addressed as shown in table 3, and documented as an AE. Infusion charts will be completed to document compliance with prescribed dose. Participants will be followed up by phone approximately 2 days after each infusion visit to check for any AEs. After the day 14 visit, no further IMP will be administered. Patients will continue their regular medications, including antithyroid drugs and betablockers throughout the study, as deemed appropriate by the clinical team.

### Assessments
Follow-up visits will take place at weeks 4, 6, 12 and 24 (±1 week); the details of assessments are provided in table 4. At each visit, concomitant medications will be recorded, an AE check will be performed, compliance with antithyroid medication checked and FBC and thyroid function (TSH, FT4, FT3 and TRAb) will be assessed.

**Table 3** Infusion-related reactions (IRRs)

| Grade 4 reactions | Life-threatening | Stop and permanently discontinue infusion. |
|---|---|---|
| Grade 3 reactions | Severe, including:<br>▶ Tachycardia ≥120 BPM or drop in SBP ≥30 mm Hg over pre-infusion with symptoms<br>▶ Rise in SBP ≥30 mm Hg over pre-infusion or tachypnoea ≥24/min<br>▶ Hypoxia SaO$_2$ <92% | Stop infusion until observations improve and restart at 50% of the infusion rate at which IRR occurred. If the patient does not experience additional symptoms, infusion rate escalation may be resumed at increments and intervals as appropriate. The procedure above should be repeated in the event of recurrence of grade 3 symptoms. After the infusion restart attempts, the IMP treatment will be permanently discontinued upon the third occurrence of a grade 3 or greater IRR. |
| Grade 1–2 reactions | Mild to moderate, including:<br>▶ nasal congestion<br>▶ cough<br>▶ chills<br>▶ throat irritation<br>▶ vomiting<br>▶ nausea | Stop infusion in case of clinically significant reaction only until observations improve and restart at 50% of the infusion rate at which IRR occurred. If the patient does not experience any further IRR symptoms, infusion rate escalation may be resumed at increments and intervals as clinically appropriate up to the maximum rate. Infusion does not need to be stopped if IRR is not clinically significant. |

BPM, beats per minute; IMP, investigational medicinal product; SaO$_2$, arterial oxygen saturation; SBP, systolic blood pressure.

Other assessments include goitre size assessment, eye examination, patient-reported quality of life measures (Graves' Orbitopathy Quality of Life Instrument[4] and Thyroid Patient-Reported Outcome[19]), specific antibody tests and other safety bloods (FBC, LFT). At the end of their participation in the trial, participants will be asked their opinion on which treatment allocation they received, and if they would recommend their treatment to a friend with the same condition. In addition, we plan to undertake exploratory analyses of lymphocyte subsets determined by flow cytometry and transcriptomic markers of plasma cells using quantitative reverse transcription-PCR.[20]

### Discontinuation of treatment and withdrawal
Participants who have completed one or more daratumumab/placebo infusions may remain on the trial but choose to discontinue treatment. These participants will continue to take part in the trial assessments as per the trial schedule of events. Alternatively, an investigator may discontinue a participant from receiving trial treatment at any time (until administration of second infusion is completed), if the investigator considers it necessary for any reason. Participants who withdraw from the trial after receiving IMP (full or part dose) will not be replaced. Participants who withdraw before IMP is given will be replaced.

### Safety reporting
AEs will be documented in patient medical notes before entry into the trial database. AEs judged by the investigator responsible for patient care as consistent with the usual clinical pattern for patients with severe Graves' disease (eg, hyperthyroidism, hypothyroidism, tachycardia, bradycardia) will not be reported. Disease progression will be recorded as an AE only if it is more severe than expected in the trial population. AEs will be coded according to MedDRA classifications. All AEs

meeting the criteria for seriousness will be reported by the investigator to the trial sponsor within 24 hours of awareness, and suspected unexpected serious adverse reactions will be reported to the Medicines and Healthcare products Regulatory Agency (MHRA) and Research Ethics Committee (REC) by the trial sponsor in line with regulatory reporting timelines.

### End of trial
The end of the trial is defined as last patient last visit. Treatment allocation will be revealed to all participants (and their general practitioner/referring clinician where applicable) by letter at the end of the trial, with contact details to discuss if required. At the end of their participation in the study, participants will return to standard NHS care.

### Payment
No payment will be made to participants for taking part in this trial, although reasonable travel expense reimbursement including taxis will be made for trial visits.

### Data management
The Clinical Data Management System (CDMS) used to capture trial data is Red Pill, supplied by Sealed Envelope. Users have individual password-protected user accounts appropriate to their trial duties (eg, randomisation, data entry, monitor). Patients are identified on the CDMS using a unique participant ID. Data monitoring will include a mixture of central, off-site and on-site approaches and will be outlined in a risk-based monitoring plan prior to the start of the trial. The trial-specific Data Management Plan will include details of how the validity and quality of data will be monitored and managed. As far as possible, validations will be built into the CDMS and additional manual validations will be detailed in the Data Validation Plan. Trial monitoring will be performed by the NCTU as

**Table 4** Schedule of assessments

| Assessment/activity | Visit 1 | Visits 2 & 3 | Visit 4 Day 0 | Visit 5 Day 14 (±3 days) | Visit 6 4 weeks (±1 week) | Visit 7 6 weeks (±1 week) | Visit 8 12 weeks (±1 week) | Visit 9 24 weeks (±1 week) |
|---|---|---|---|---|---|---|---|---|
| Written informed consent | X | | | | | | | |
| Demographics & medical history | X | | | | | | | |
| Physical examination | X | | | | | | | |
| 12-lead ECG | X | | | | | | | |
| Eye examination (CAS) | X | | | | | | | |
| Hepatitis B & C serology, HIV | X | | | | | | | |
| Bone chemistry | X | | | | | | | |
| Coombs, blood group, RBC phenotyping | X | | | | | | | |
| Pregnancy testing (females) | X | | | | | | X | |
| COVID-19 vaccination offered | | X X | | | | | | |
| IMP infusion | | | X | X | | | | |
| Weight | X | X | X | X | X | X | X | X |
| Resting pulse rate | X | | X | X | X | X | X | X |
| Goitre size assessment | X | | X | | | X | X | X |
| FBC | X | | X | X | X | X | X | X |
| TSH, FT4, FT3, TRAb | X | | X | X | X | X | X | X |
| U&E, LFT | X | | X | X | X | X | | |
| IgGAM | X | | X | | | X | X | X |
| Height | | | X | | | | | |
| Eye examination (composite eye index) | | | X | | | X | X | X |
| Thyroid volume on USS | | | X | | | | | X |
| QoL questionnaires (GOQoL, ThyPRO) | | | X | | | X | X | X |
| TgAb, ATPO | | | X | | | X | X | X |
| Specific antibodies | | | X | | | | X | X |
| Exploratory analysis sample | | | X | | | X | X | X |
| Vital signs | | | X | X | | | | |
| Compliance with ATD | | | X | X | X | X | X | X |
| Concomitant medication | | X | X | X | X | X | X | X |
| Adverse event reporting | | X | X | X | X | X | X | X |
| Follow-up phone call | | | X | X | | | | |
| End of trial questions | | | | | | | | X |

ATD, antithyroid drug; ATPO, antithyroid peroxidase; CAS, Clinical Activity Score; FBC, full blood count; GOQoL, Graves' Orbitopathy Quality of Life Instrument; IMP, investigational medicinal product; LFT, liver function test; QoL, quality of life; RBC, red blood cell; TgAb, thyroglobulin antibodies; ThyPRO, Thyroid Patient-Reported Outcome; TRAb, TSH receptor antibodies; TSH, thyroid-stimulating hormone; U&E, urea and electrolytes; USS, ultrasound scan.

per the monitoring plan and the trial is also liable to audit by the trial sponsor and regulator.

Until publication of the trial results, access to the full dataset will be restricted to the Trial Management Group (TMG) and to authors of the publication. Requests for anonymised data should be directed to the lead author.

## Analysis

The primary analysis will assess whether there is a significant dose–response using percentage reduction in serum TRAb concentration from baseline to 12 weeks using the intention-to-treat population (participants analysed according to randomisation allocation).

To do this, we will calculate a contrast-based test statistic at a one-sided 5% type I error rate.[21] If this test statistic is significant, we will test whether there is a mean difference in the outcome between individual doses and placebo using a suitable linear model. We will also perform a simple test of all placebo participants versus all daratumumab patients using a linear model that adjusts for presence of eye disease. We will also fit quadratic and three-parameter Emax models to assess dose–response.[22] This analysis will

inform selection of the most suitable dose for subsequent trials. Exploratory analyses using subgroups of patients including by gender, smoking status, presence of TED and change in serum TRAb will be undertaken.

### Interim analyses and criteria for the premature termination of the trial

An interim analysis of dose–response will be conducted after 15 patients have provided 12-week follow-up data. To assess efficacy, a three-parameter Emax model will be fitted to the logarithm of the percentage change in TRAb from baseline to week 12 using the DoseFinding package in R[23 24] in the dose–response population (participants analysed according to the amount of daratumumab received). This model is used to estimate the plateau effect and (1) the dose that gives a mean reduction of 50% and (2) the dose that gives 90% of the plateau effect. If the estimated plateau effect corresponds to >50% reduction in TRAb, the trial will continue with placebo and doses (1) and (2) defined above. If the estimated plateau corresponds to less than 25% reduction in TRAb, the trial will terminate early for lack of promising dose–response unless the Data Monitoring Committee (DMC) agrees that there has been an unequivocal improvement in serum thyroid hormone measurements in the active IMP versus placebo groups; if it is between 25% and 50% reduction, the trial will continue with doses of 9 mg/kg and a higher daratumumab dose (eg, 12 or 16 mg/kg as determined by safety data), plus placebo. Safety information will be taken into consideration in deciding stage 2 dose(s) and could override the model-based dose recommendations.

### Statistical size calculations

A 50% or more reduction in TRAb concentration from baseline within 12 weeks of intervention was chosen as biologically plausible and clinically important. An observational study of patients with milder Graves' disease showed a mean change in TRAb concentration of 2% over 6 weeks, with a standard deviation (SD) of 15%. Assuming an SD of 25%, to account for the outcome being measured at 12 weeks, a 5% reduction in placebo-treated subjects, an Emax dose–response, a plateau 60% reduction in TRAb concentration over 12 weeks with active IMP and 1 mg/kg dose giving half of the total, 27 participants will give over 90% power (5% one-sided type I error) to conclude there is a dose–response relationship.

In order to allow for 10% dropout, we will plan to recruit 30 participants.

Power was verified through simulations in 'R',[24] assuming the logarithm of the percentage reduction to be normally distributed with SD of 0.5 (equivalent to 25% SD when the mean is 50% reduction). The power/type I error rate for a range of scenarios is given in table 5.

### Trial oversight

The TMG will meet approximately monthly throughout the trial, and includes the Chief Investigator, statisticians, NCTU staff, pharmacy and sponsor representatives. Oversight is provided by an independent DMC that will meet at the start of the trial, at the interim analysis point and on an ad hoc basis throughout, as well as a Trial Steering Committee (TSC) that will generally meet following a DMC meeting. The terms of reference and composition of these committees are outlined in their respective charters.

### Protocol amendments

NCTU will submit amendments to the Health Research Authority and MHRA/REC (as appropriate) as well as informing ISRCTN, trial recruiters, investigators and the relevant research and development department of changes. A list of version changes will be kept in the trial protocol appendix and can be found in online supplemental material 2.

### Trial status

This manuscript is based on protocol V.6.0 dated 18 May 2023. The Graves-PCD trial opened to recruitment on 29 September 2021. At the time of writing, stage 2 of the trial is underway.

### Patient and public involvement

Patients were involved in the design of the trial via the BTF (https://www.btf-thyroid.org/), and lay members will have oversight and input during the trial and dissemination as members of the TSC.

### ETHICS AND DISSEMINATION

The trial received a favourable ethical opinion from London-Hampstead REC (reference 21/LO/0449). The results of this trial will be disseminated at international

**Table 5** Power/type 1 error rate calculations

| Scenario | Probability to conclude dose–response (10 000 replicates) |
|---|---|
| 1—plateau of 80% reduction ED50=3 mg/kg | >99% |
| 2—plateau of 60% reduction, ED50=1 mg/kg | 92% |
| 3—plateau of 60% reduction, ED50=3 mg/kg | 83% |
| 4—null scenario | 3.2% |

The R code required to reproduce these results can be found in online supplemental material 1.
ED50, median effective dose.

meetings, in the peer-reviewed literature and through partner patient group newsletters and presentations at patient education events.

**Acknowledgements** We would like to acknowledge the ongoing support provided by the British Thyroid Foundation. We are grateful to Professor Ruth Andrew of Edinburgh University for advice concerning dose choices and to Dr Catherine Stroud of the Newcastle upon Tyne Hospitals NHS Foundation Trust for immunological safety advice.

**Contributors** SHP had the original idea for the trial. The trial protocol was developed by SHP, with substantial input and revision by FW, JWas, MC, RHM, JWal, LG and PB. IB and NM also made changes to the protocol at various stages of development. The trial protocol manuscript was primarily drafted by FW, with substantial input from SHP and JWas. All other coauthors approved the final version of the manuscript.

**Funding** This trial is funded by the UKRI Medical Research Council (MRC; ref: MR/V005898/1).

**Competing interests** SHP has received speaker fees from Merck and IBSA, and consulted for Apitope/Worg and Immunovant/Roivant. JWas has consulted for Worg.

**Patient and public involvement** Patients and/or the public were involved in the design, or conduct, or reporting, or dissemination plans of this research. Refer to the Methods section for further details.

**Patient consent for publication** Not applicable.

**Provenance and peer review** Not commissioned; externally peer reviewed.

**ORCID iDs**
Faye Wolstenhulme http://orcid.org/0000-0002-4086-7931
Irena Bibby http://orcid.org/0009-0007-3105-9864
Rebecca H Maier http://orcid.org/0000-0002-7350-3288
James Wason http://orcid.org/0000-0002-4691-126X

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
