## [Reviewer comments · BMJ Open]

ARTICLE DETAILS

TITLE (PROVISIONAL)	Graves-PCD: Protocol for a randomised, dose-finding, adaptive trial of the plasma cell depleting agent daratumumab in severe Graves' disease
AUTHORS	Wolstenhulme, Faye; Bibby, Irena; Cole, Michael; Gixti, Lydia; McGregor, Naomi; Bradley, Penny; Maier, Rebecca; Walker, Jenn; Pearce, Simon H; Wason, James

VERSION 1 – REVIEW

REVIEWER	Yuan, HuiJuan Institute of People's Hospital of Zhengzhou University, Department of Endocrinology and Metabolism
REVIEW RETURNED	29-Oct-2023

GENERAL COMMENTS	This study investigated the efficacy of daratumumab in patients with severe Graves' disease and provided new options for patients that we believe will attract the interest of readers 1、 What's the basis for the use of drug concentration and concentration gradient in the article? Please give more detailed explanation.2、 In stage 1, three persons were included in each subgroup. At the same time, the matching of age and gender between groups should be considered. An expanded sample size is also necessary.3、 The trial design should be more detailed and specific.
---

REVIEWER	Antonelli, Alessandro University of Pisa, Clinical and experimental medicine I am Consultant for Horizon Therapeutics Italy S.r.l.
REVIEW RETURNED	02-Nov-2023

GENERAL COMMENTS	The study protocol is interesting and well-described. I suggest to publish it in this version.
--

REVIEWER	Glushkova, Natalya Semey Medical University, Department of Epidemiology, Evidence Based Medicine and Biostatistics
REVIEW RETURNED	06-Nov-2023

GENERAL COMMENTS	The manuscript is both important and interesting. The topic holds significant relevance in the field. However, several questions were raised during the review process. On the registration page (https://www.isrctn.com/ISRCTN81162400) for the trial, it indicates that the study lasts from February 2021 to January 2024. Is this study still ongoing? The protocol manuscript should only report planned or ongoing studies. The primary outcome in the manuscript is mentioned as the change in serum TRAb antibodies from baseline to 12 weeks. Yet, in the limitations, it's stated that "the long-term remission of hyperthyroidism cannot be assessed in the 24-week follow-up period". Both outcomes might be considered secondary in terms of clinically significant outcomes such as remission. The study seems to measure surrogate endpoints. Table 1 does not define abbreviations in the footnotes: FACS, GOQoL, etc. How can the division into five groups of n=3, randomized at a 1:1:1:1:1 ratio, be justified as adequate, even for an interim clinical study? What are the chances that the results could be biased? Line 46 mentions 30 adult patients, but later only 15 are referenced in line 158. On the registration page (https://www.isrctn.com/ISRCTN81162400), 30 patients are also mentioned. How many groups and how many patients were in the 1st stage? Lines 163-164 mention that the remaining patients will be recruited in stage 2. Does this mean the remaining 15 out of the planned 30 patients? The risk assessment section does not provide any information about the risks exceeding those of routine intravenous infusions for GD patients. Are the number of IV infusions typically the same? It would be more informative if a reference to the statistical method used for sample size calculation was added to the "Statistical Size Calculations" section.
--

REVIEWER	Fisher, Benjamin
	University of Birmingham, Institute of Inflammation and Ageing
REVIEW RETURNED	29-Jan-2024

GENERAL COMMENTS	This is a clearly written trial protocol for an ongoing clinical trial of a plasma cell depleting drug in severe Grave's disease. I have a couple of minor points only:  1. Exclusion criteria specify exclusion for Hepatitis B, although the blood tests only include HbSAg. How do the authors handle patients who are isolated core antibody positive? Or is this not measured? 2. In the Table 4 schedule of assessments, it is unclear to me what the purpose of visits 2 and 3 is and it would be helpful to elaborate in a footnote. Although this is a schedule of assessments, it may be helpful to add IMP administration 3. The protocol paper is based upon version 6.0 of the protocol. It would be useful to include a table detailing protocol amendments and their rationale.
--

VERSION 1 – AUTHOR RESPONSE

Reviewer: 1

Prof. HuiJuan Yuan, Institute of People's Hospital of Zhengzhou University

Comments to the Author:

This study investigated the efficacy of daratumumab in patients with severe Graves' disease and provided new options for patients that we believe will attract the interest of readers

1、 What's the basis for the use of drug concentration and concentration gradient in the article?

Please give more detailed explanation.

A: Thank you for the interesting question. Daratumumab is used routinely in myeloma at 16mg/Kg every 2-4 weeks, and early-phase studies showed that 8mg/Kg had similar efficacy to 16mg/Kg. We felt that lower doses could likely have efficacy in Graves' disease, which is caused by one or more benign clones of autoreactive plasma cells, rather than malignant plasma cells in myeloma. A "rule of thirds" is frequently used in pharmacology studies to look at a range of doses, so we took 9mg/Kg as likely close to the top of our dose range and used a rule of thirds for the other doses. This was detailed in the manuscript, lines 120-129.

2、 In stage 1, three persons were included in each subgroup. At the same time, the matching of age and gender between groups should be considered. An expanded sample size is also necessary.

A: The analysis using a dose-response curve means that patients at the higher end of the administered dose and those at the lower end can all contribute to the information about possible efficacy at more than one dose; That is 6 patients will receive 0.5 or 1mg/Kg dosage and these can all inform our understanding of the response to lower doses using the dose-response curve approach. This improves the power as our analysis does not take data from each group of 3 participants in isolation. The methodological detail of the dose-response analysis is explained in line 394-403 and 408-413 of the MS.

As this is an early-phase study, looking only for a signal of efficacy, age and gender matching was not felt to be practical given the small sample sizes in each group.

3、 The trial design should be more detailed and specific.

A: Thank you. We have added an additional figure (figure 1) to detail the design and a full catalogue of scheduled trial events are detailed in an updated table 4. Much of the design section is taken directly from our protocol document and accords with the SPIRIT recommendations for protocol documentation. Please elaborate where you feel other details or specificity is lacking and we will be happy to provide it.

Reviewer: 2

Dr. Alessandro Antonelli, University of Pisa

Comments to the Author:

The study protocol is interesting and well-described. I suggest to publish it in this version.

A: Thank you for this positive evaluation.

Reviewer: 3

Dr. Natalya Glushkova, Semey Medical University

Comments to the Author:

The manuscript is both important and interesting. The topic holds significant relevance in the field.

However, several questions were raised during the review process.

On the registration page (<https://www.isrctn.com/ISRCTN81162400>) for the trial, it indicates that the study lasts from February 2021 to January 2024. Is this study still ongoing? The protocol manuscript should only report planned or ongoing studies.

A: Thank you. The trial start was delayed by COVID safety and staffing issues until October 2021, and the trial will not finish until April 2024. These details have been added to line 160&161, and the ISRCTN record has been updated.

The primary outcome in the manuscript is mentioned as the change in serum TRAb antibodies from baseline to 12 weeks. Yet, in the limitations, it's stated that "the long-term remission of hyperthyroidism cannot be assessed in the 24-week follow-up period". Both outcomes might be considered secondary in terms of clinically significant outcomes such as remission. The study seems to measure surrogate endpoints.

A: We agree with this: the trial measures TRAb concentration as a surrogate endpoint for long-term outcome of Graves' disease. This is an experimental medicine trial looking to repurpose an existing drug for a novel indication, so at this stage (the first ever Graves' patients treated with this compound) we are looking simply for a signal of efficacy and an idea of a potentially efficacious dose, to give us the evidence to go forward to a larger proof of principle study. It would not be justified to perform a longer study or deny effective conventional therapies for longer until there was some indication of efficacy.

Table 1 does not define abbreviations in the footnotes: FACS, GOQoL, etc.

A: Thank you, this has been amended, Line 151-153.

How can the division into five groups of n=3, randomized at a 1:1:1:1:1 ratio, be justified as adequate, even for an interim clinical study? What are the chances that the results could be biased?

A: Thank you. As a dose-response curve analysis will be performed, our analysis does not take data from each group of 3 participants in isolation. This is also explained in more detail in our response to reviewer 1's second question, above.

Line 46 mentions 30 adult patients, but later only 15 are referenced in line 158. On the registration page (<https://www.isrctn.com/ISRCTN81162400>), 30 patients are also mentioned. How many groups and how many patients were in the 1st stage? Lines 163-164 mention that the remaining patients will be recruited in stage 2. Does this mean the remaining 15 out of the planned 30 patients?

A: Yes, exactly. There will be 15 patients in stage 1 and 15 in stage 2. We have clarified this on lines 167 and 170 and by the addition of figure 1.

The risk assessment section does not provide any information about the risks exceeding those of routine intravenous infusions for GD patients. Are the number of IV infusions typically the same?

A: It is unclear exactly what the reviewer is asking. Each patient in the study will receive two IV infusions either of daratumumab or placebo (the same number). This detail has been added to the schedule of assessments table (table 4) to make it clearer. Conventional treatment of Graves' disease would not involve any infusions.

It would be more informative if a reference to the statistical method used for sample size calculation was added to the "Statistical Size Calculations" section.

A: Thank you. The sample size calculation was coded in the statistics environment 'R'. The generic citation for 'R' has been included in the reference list (reference #13) and the R code used has been included as supplementary material.

Reviewer: 4

Dr. Benjamin Fisher, University of Birmingham

Comments to the Author:

This is a clearly written trial protocol for an ongoing clinical trial of a plasma cell depleting drug in severe Grave's disease. I have a couple of minor points only:

1. Exclusion criteria specify exclusion for Hepatitis B, although the blood tests only include HbSAg. How do the authors handle patients who are isolated core antibody positive? Or is this not measured?

A: Thank you for this pertinent point. Full hepatitis B serology will be performed and core antibody positivity could mean active infection so such patients will be excluded. Table 4 has been amended to clarify this point (HbSAg was used as shorthand in the table for hepatitis B testing).

2. In the Table 4 schedule of assessments, it is unclear to me what the purpose of visits 2 and 3 is and it would be helpful to elaborate in a footnote. Although this is a schedule of assessments, it may be helpful to add IMP administration

A: Thank you for this comment. At the time of writing the protocol (summer-autumn 2020), many patients had not yet been offered routine COVID-19 vaccinations (eg. they were young), so visits 2 and 3 were to offer them COVID vaccinations prior to the trial. IMP administration has also been added to the table as requested.

3. The protocol paper is based upon version 6.0 of the protocol. It would be useful to include a table detailing protocol amendments and their rationale.

A: Thank you. A table of major protocol amendments has been added as a supplementary document.

VERSION 2 – REVIEW

REVIEWER	Yuan, HuiJuan Institute of People's Hospital of Zhengzhou University, Department of Endocrinology and Metabolism
REVIEW RETURNED	09-May-2024

GENERAL COMMENTS	The overall level of the paper is good: it is well written, and some important considerations are highlighted. I am very glad the authors wrote this essay. I really appreciate what this research group has tried to do, it is a very interesting problem. It will be better to edit the grammar and syntax.
---

REVIEWER	Glushkova, Natalya Semey Medical University, Department of Epidemiology, Evidence Based Medicine and Biostatistics
REVIEW RETURNED	20-Mar-2024

GENERAL COMMENTS	The manuscript is intriguing and very actual to current discussions. After a thorough review, I have several comments, primarily concerning technical aspects: 1. Lines 50 and 149: The primary outcome mentioned is the change in serum TRAb antibodies from baseline to 12 weeks. It would be beneficial to clarify whether this change is expressed as a percentage or if it denotes a statistically significant difference from the baseline level.2. Line 97: The reference style used from this line onwards differs from that used previously.3. Has the study been registered in any publicly available databases, such as clinicaltrials.gov? Providing registration information enhances transparency and facilitates tracking of the study's progress and findings.
---

REVIEWER	Fisher, Benjamin University of Birmingham, Institute of Inflammation and Ageing I have undertaken consultancy for Novartis, BMS, Servier, Galapagos, Roche, UCB, Sanofi, Janssen and received research funding from Janssen, Servier, Galapagos, Celgene. None of these relationships are directly related to the topic of the manuscript.
REVIEW RETURNED	13-Mar-2024

GENERAL COMMENTS	Thank you. My comments have been addressed.
---

VERSION 2 – AUTHOR RESPONSE

Thank you for this opportunity to submit a minor revision, please find our responses below.

1. Lines 50 and 149: The primary outcome mentioned is the change in serum TRAb antibodies from baseline to 12 weeks. It would be beneficial to clarify whether this change is expressed as a percentage or if it denotes a statistically significant difference from the baseline level.

It has been clarified in these lines as well as line 411, that this is a percentage.

2. Line 97: The reference style used from this line onwards differs from that used previously.

Thank you for flagging this, there were some references that hadn't pulled through correctly from our reference software, these have now been updated.

3. Has the study been registered in any publicly available databases, such as clinicaltrials.gov? Providing registration information enhances transparency and facilitates tracking of the study's progress and findings.

Yes, the trial was prospectively registered on ISRCTN and the registration number is provided on line 58.